# Prospective Analysis of Functional and Structural Changes in Patients with Spinal Muscular Atrophy—A Pilot Study

**DOI:** 10.3390/biomedicines10123187

**Published:** 2022-12-08

**Authors:** Aleksandra Bieniaszewska, Magdalena Sobieska, Ewa Gajewska

**Affiliations:** 1Department of Developmental Neurology, Poznan University of Medical Sciences, 60-355 Poznan, Poland; 2Doctoral School, Poznan University of Medical Sciences, 60-812 Poznan, Poland; 3Department of Rehabilitation and Physiotherapy, Poznan University of Medical Sciences, 61-545 Poznan, Poland

**Keywords:** spinal muscular atrophy, functional assessment, functional scales, ranges of motion, contractures

## Abstract

Spinal muscular atrophy (SMA) is a rare, autosomal recessive neuromuscular disease. Recent years have seen a significant development of therapeutic options for SMA patients. With the development of treatment methods, it has become necessary to adapt a physiotherapeutic approach to the evolving clinical picture of SMA patients. We presented an analysis of 40 SMA patients undergoing pharmacological treatment, examined twice in an average interval of 5 months. Twelve patients (non-sitters) were evaluated using CHOP-INTEND, while 28 (sitters) were tested using the Hammersmith scale. The research protocol consisted of measurements of upper and lower limb ranges of motion, and four tests for early detection of musculoskeletal changes. Both non-sitters and sitters patients showed motor improvement between the first and second examinations. Favorable changes in range of motion parameters were noted in most children, except for hip extension (HE) range, which deteriorated. An association was also observed between scale scores and the presence of contractures in the hip and knee joints depending on the group studied. Our findings showed that the presence of contractures at the hip and knee joint negatively affected functional improvement as measured by the scale scores.

## 1. Introduction

Spinal muscular atrophy (SMA) is a rare neuromuscular disease, inherited in an autosomal recessive manner and characterized by degeneration of alpha motoneurons in the spinal cord and brainstem [1,2]. In 95% of cases, SMA is caused by deletion and/or mutation in the survival motor neuron 1 *(SMN1)* gene on chromosome 5, which results in insufficient production of the survival motor neuron (SMN) protein [1,3]. It manifests as muscle flaccidity and gradual atrophy, resulting in progressing functional limitations [3,4]. Motor development is known to regress in untreated patients. Characteristics symptoms of SMA include the loss of previously acquired motor function in children, with the addition of respiratory distress, dysphagia, and joint contractures [2,5]. From 2005 to 2015, the incidence of SMA in Poland was estimated at 10.3–13.5/100,000 live births, while the average incidence in Europe from 2011 to 2015 was 11.9/100,000 [6,7]. The Polish SMA Registry reports 720 persons with spinal muscular atrophy as of 1 January 2020. However, it is estimated that around 1000 persons are affected [8].

The classification distinguishes types of SMA based on the child’s age at the onset of the first symptoms and the highest motor function achieved. Type 1 (SMA1; Werdnig-Hoffmann) applies to patients who never achieve the function of sitting up on their own, with the onset of symptoms occurring in the first few months of life. Type 2 (SMA2) are sedentary patients with a diagnosis between 6 and 18 months of age, who will never achieve independent walking, while mild type 3 (SMA3; Kugelberg-Welander) applies to children whose highest function is independent walking, and the onset of symptoms occurs after 18 months of age [4,7,9].

Recent years have seen a significant development of therapeutic options for SMA patients. Knowledge of the causes of SMA has provided the opportunity to develop a targeted treatment based on increasing the level of SMN protein [10]. As of 1 January 2019, reimbursement of Nusinersen (Spinraza^®^)—an antisense oligonucleotide that modifies pre-mRNA splicing of *SMN2* gene to promote increased production of full-length SMN protein, has been introduced in Poland [3,11]. At the same time, as part of a global access program and through clinical trials, some patients are able to receive Risdiplam (Evrysdi^®^)—an *SMN2*-directed RNA splicing modifier increasing the ability of SMN2 to produce full-length SMN protein [8,12,13]. Furthermore, some children are participating in gene therapy with the help of private funding. Zolgensma (Onasemnogene abeparvovec) uses adenovirus serotype 9 (AAV9), which is a vector to deliver a synthetic DNA sequence corresponding to the *SMN1* gene to motor neurons, resulting in the formation of the missing SMN protein [9,10,14]. From 1 September 2022, reimbursement of all these drugs was introduced in Poland. Modern therapeutic approaches have significantly improved the quality of life of patients with SMA. With treatment increasing the production of SMN protein, infants and children reach new developmental milestones that were previously impossible. Early implementation of therapy results in, for example, the ability of SMA1 patients to sit up independently and sometimes even achieving standing and walking function [3,13,14]. The differences were also observed in adolescents whose treatment influenced improvement or stabilization of motor function. However, greater improvement was seen in younger children [12,15].

All of the patients with SMA in Poland are diagnosed and rehabilitated according to the standards based on the International Standards of Care Committee for Spinal Muscular Atrophy created by experts in 2007 [16]. This document was updated in 2018 and includes new standards based on the experience of the last decade and prepared by a multidisciplinary team of a dozen experts from around the world (including Poland). These publications are available on the SMA Foundation website and cover such issues as diagnostic, rehabilitation, orthopaedic care, nutrition, pulmonary and intensive care, treatment, supplementation and vaccination, other organs involvement and ethics [8,17,18]. Moreover, these standards consider care by an occupational therapy, speech therapy and alternative communication specialist, anesthesiology issues, as well as equipment and assistive technology issues [17,18].

As treatment methods evolve, it is necessary to adapt physiotherapeutic approaches to the evolving clinical picture of SMA patients. Observing the relationship between the development of joint contractures and the improvement/deterioration of specific motor functions is essential to adjust the provided physiotherapy. Guidelines using validated diagnostic tools, such as functional scales, goniometer, scoliometer and plurimeter have become the standard for monitoring the course of the disease [11,19,20].

The functional scales, such as: Children’s Hospital of Philadelphia Infant Test of Neuromuscular Disorders (CHOP-INTEND), Hammersmith Functional Motor Scale—Expanded (HFMSE), Hammersmith Infant Neurological Examination (HINE) and Motor Function Measure (MFM), are used to assess children’s motor skill progress [21,22,23,24]. In turn, the Revised Upper Limb Module (RULM) scale is used to assess upper limb performance, and the 6 min walk test is used to evaluate the walking ability of patients with SMA [25,26]. In the present study, the CHOP-INTEND and HFMSE scales were used to assess motor development.

The CHOP-INTEND scale is used to functionally assess patients with the most severe form of SMA, who are not able to sit unsupported. It is based on the evaluation of 16 activities performed by the child during purposeful movements or during spontaneous activity [21]. In turn, the HFMSE scale is a diagnostic tool used to screen patients with SMA2 and SMA3, who are able to sit unsupported. The HFMSE scale consists of a basic version of 20 questions, assessing activities such as prolonged sitting, changing position from sitting to lying, crawling or standing [22]. The scale also includes a supplementary module of 13 questions derived from the Gross Motor Function Measure (GMFM) scale, assessing motor activities such as squats, forward jumps, and going upstairs and downstairs with and without assistance [27].

To accurately assess the condition of patients with spinal muscular atrophy, measurements of upper and lower limb joint mobility are taken using a goniometer to determine the neutral position of the limbs and the passive range of motion, together with measurements of cervical mobility, and pelvic or spine alignment [19,20].

The current literature shows only one publication about the relationship between structural changes and motor function in children with SMA undergoing pharmacological treatment, which we believe affects physiotherapeutic assessment [28]. Moreover, previous publications presented functional improvement based only on total scale results, which is one of the limitations of the scales used in clinical practice. Furthermore, physiotherapeutic assessment according to these functional scales requires extensive clinical experience in SMA patients and thorough training of the examiner [3,29,30]. However, there is a lack of data on whether the functional improvement is related to all motor skills or perhaps only selected functions and the relationship of the improvement to contracture formation.

The purpose of this study is a prospective analysis of motor progression and structural changes in patients with spinal muscular atrophy undergoing pharmacological treatment and physiotherapy, shown from the perspective of physiotherapists.

## 2. Materials and Methods

This prospective study used data collected between May 2020 and February 2022. All patients were assessed at the time of enrollment and followed up for five months.

### 2.1. Materials

Forty patients (22 females and 18 males) aged 4 months to 20 years (Me = 7.5 years, Q25–Q75 = 5–12) of the Department of Developmental Neurology and outpatient clinics from the Wielkopolska region, were recruited for the study. The patients enrolled in the study were treated with Nusinersen or Risdiplam or underwent genetic treatment. All patients also received appropriate physiotherapy. The duration between the start of treatment to the first examination was around 15 months (7–32). Detailed characteristics of the study groups are included in Table 1. The exclusion criteria comprised the presence of other genetic diseases affecting the child’s motor development, as well as lack of parental consent for the child to participate in the study.

### 2.2. Methods

Patients were evaluated twice with an average interval of 5 months between the examinations. The research protocol consisted of functional assessment of gross motor skills and measurement of upper and lower limb ranges of motion, as well as four tests for early detection of musculoskeletal changes previously developed and described in the literature [19]. In addition, at the first examination, parents completed a personal questionnaire regarding diagnosis and the child’s age of its setting, the number of SMN gene copies (confirmed by medical history), the type of treatment received, child’s functional level, and the type and intensity of rehabilitation provided.

The CHOP-INTEND and HFMSE functional scales were used to assess motor function. The CHOP-INTEND scale was used to examine non-sitters’ children with SMA1. According to the scale, the following movements were assessed: spontaneous movements (upper extremity), spontaneous movement (lower extremities), hand grip, head in midline with visual stimulation, hip adductors, rolling: elicited from legs, rolling: elicited from arms, shoulder and elbow flexion and horizontal abduction, shoulder flexion and elbow flexion, knee extension, hip flexion and foot dorsiflexion, head control, elbow flexion, neck flexion, Landau, Galant. The maximum number of points on the scale is 64, with each task scored from 0 to 4 points [21]. The sitters group, children with SMA2 and SMA3, were examined with the HFMSE scale. The quality of the following motor functions/activities was assessed: plinth/chair sitting, long sitting, one/two hands to head in sitting, supine to side lying, rolling prone to supine, rolling supine to prone, sitting to lying, propping on forearms, lifting head from prone, propping on extended arms, lying to sitting, four-point kneeling, crawling, lifting head from supine, supported standing, unsupported standing, stepping, right and left hip flexion in supine, high kneeling to half kneeling, high kneeling to standing, stand to sitting on the floor, squat, jumping forward, and ascending and descending stairs (with railing and without arm support). Each question can be scored from 0 to 2 points, where 2 points means the task was completed correctly, 1 means the task was completed with compensation, and 0 means the task was not completed. The maximum number of points possible to obtain is 66 [22,27].

A physiotherapist, who used the CHOP-INTEND or HFMSE scales, had previously been trained and participated in annual reminder meetings. A manual containing all the procedures for applying the scales was used during the assessment. All items were tested without using a brace or orthoses [21,22,27,28].

Upper and lower limb neutral position and ranges of motion were assessed in all patients using a goniometer [20]. If the child was unable to reach a given position, the value of joint contracture was determined. The evaluated parameters included shoulder flexion (SF) and abduction (AbS), elbow flexion (EF), forearm pronation (PF) and supination (SF) in the upper limb, as well as hip flexion (HF), knee flexion (KF) and ankle dorsiflexion (AD—ankle dorsiflexion) in the lower limb. In addition, four parameters for detecting early musculoskeletal changes that predispose to scoliosis formation were examined [4,19,28]. The supine angle of trunk test (SATR), measuring the angle of upper (SATR-U) and lower (SATR-L) trunk rotation, and pelvic obliquity (PO), determining the value of pelvic tilt angle, were examined using a scoliometer. Other parameters included cervical rotation test (CR), presenting the extent of cervical spine rotation, and hip extension (HE). Both parameters were measured using a plurimeter [4,19,28].

The approval of the Poznan University of Medical Sciences Bioethical Committee was obtained for conducting the study (no. 1035/19).

### 2.3. Statistical Analysis

All results were analyzed using Statistica 12.2 Software by StatSoft (Krakow, Poland). As the distribution of data for interval variables differed from normal (Shapiro–Wilk and Lilleforse tests), and for all ordinal variables, the results were shown as median and quartiles [Me (Q25–Q75)] for all variables. Non-parametric statistical tests were used, i.e., Mann–Whitney U test to assess the differences between the two groups and the sign test to assess the difference in time. Spearman’s rank correlation (rho) was used to calculate the correlation between the variables. For time variability, the sign test was used. In all tests, the results were considered significant if *p* < 0.05. All calculations were performed using Statistica.pl.

## 3. Results

A total of 40 children participated in the study. Each child had a genetically confirmed diagnosis of SMA1 (*n* = 14), SMA2 (*n* = 18) or SMA3 (*n* = 8). Participants were divided into two groups based on their ability to sit. Twelve children (non-sitters group) were examined using the CHOP-INTEND scale, while twenty-eight (sitters group) were assessed using the Hammersmith Functional Motor Scale—Expanded. The duration between the diagnosis to the start of the treatment was around 48 months (25–111). There was no significant difference between the sitters and non-sitters groups (z = 1.47; *p* = 0.142). Detailed characteristics of the study groups is included in Table 1.

Median result for the non-sitters group, assessed with CHOP-INTEND scale was 22 (Q25–Q75 = 7–37; min–max = 2–52). In the second examination, this group achieved a median of 23 points (Q25–Q75 = 10–41; min–max = 2–60). In turn, for the sitters group, the first examination with HFMSE resulted in 22 points for the first part (Q25–Q75 = 14–32; min–max = 3–40) and 24 points for the complete scale (Q25–Q75 = 16–36; min–max = 5–63). In the second examination, the first part and complete scale scores were 26 (Q25–Q75 = 14–33; min–max = 4–40) and 30 points (Q25–Q75 = 16–38; min–max = 6–64), respectively. Thus, the median for patients in the non-sitters group increased by 1 point during 5-months follow-up, while patients in the sitters group improved by 4 points in the first part and 6 points from the total functional scale score. Such a high score resulted from significant improvements in two children. The first patient improved by 13 points and started treatment at the time of the first examination. The improved motor function without compensation was long sitting, one/two hands to head in sitting, propping on forearms, lying to sitting, lifting head from supine, right and left hip flexion in supine. The other child improved by 9 points and started treatment in the first month of life with our first follow-up time after 24 months. The improved motor function was lifting head from prone, four-point kneeling and crawling (without compensation) and sitting to lying, lying to sitting and high kneeling to half kneeling (with compensations). In some items of the CHOP-INTEND and HFMSE scales, significant changes can be observed, as presented in Figure 1 and Figure 2.

The correlations between *SMN2* copy numbers and clinical scores were also analyzed. There were no statistically significant correlations between the number of gene copies and the result on CHOP-INTEND nor HMFSE scales.

Improvement in the overall CHOP-INTEND score mainly relates to spontaneous lower limb movement (7 patients), head control (3 patients), hip flexion (3 patients), and elbow flexion (3 patients). Lower scores are observed mainly in functions such as keeping the head centered and horizontal abduction and affected 2 patients (Figure 1).

The main items that improved the HFMSE score included rotation from abdomen to back over the left shoulder (5 patients), change in position from prone to sitting (5 patients), and flexion of the left hip (6 patients). Most of the parameters in both HFMSE and CHOP-INTEND scales remained unchanged, which is considered clinically favorable in SMA patients (Figure 2).

When children from the sitters group were divided according to the drug they received (Nusinersen versus Risdiplan), no statistically significant differences were found in HFMSE in partial score nor total score, both at the beginning or at the end of the observation period. *p*-values obtained were as follows: 0.119; 0.115; 0.094; 0.104. In addition, the number of children who showed unchanged result, improvement or deterioration did not differ significantly between the groups divided according to the drug used (*p* = 0.603).

The next step of the study was to analyze the changes in mobility range parameters between the 1st and 2nd examinations, as well as estimate the statistical significance of these changes. The median (Q25–Q75) and min-max ranges were reported, together with the significance of the difference, calculated using the sign test. The z- and *p*-values are presented in Table 2.

A slight increase in the range of left and right hip flexion (HF-L and HF-R) and left knee flexion (KF-L) was observed in all subjects. There was also improvement in cervical spine rotation (CR-L) mainly to the left and worsening of left and right hip extension (HE-L and HE-R) measured by plurimeter. The lack of significance may be due to the fact that some subjects showed improvement while others showed worsening or no change (SATR-U). To illustrate this, individual data are provided in Table 3.

The table does not include data on ankle dorsiflexion (AD) and upper limb measurements: shoulder flexion (SF) and abduction (AbS), elbow flexion (EF) and pronation (PF) and supination (SF) of the forearm, because in all examined children the ranges did not differ from the norm.

Improvement was shown in almost all parameters. The greatest change was observed in left and right hip flexion (HF-L; HF-R). Right knee flexion (KF-R), pelvic angle (PO-Angle), as well as upper (SATR-U) and lower (SATR-L) trunk rotation angles showed no change between the first and second examinations. Interestingly, in the case of KF-R and SATR-L parameters, the lack of change was observed in the sitters group, while the non-sitters group showed a general improvement. In contrast, cervical spine rotation improved in the whole group and the sitters subgroup. Hip extension parameters (HE-L and HE-R) are the only ones presenting deterioration. These changes for the left hip concern patients in both groups, while changes for the right hip concern the whole group and the sitters subgroup. The difference in scores achieved, with patients divided according to the presence of joint contracture at the first examination, was then estimated along with significance estimation (Table 4).

The remaining joints were not included in Table 4 as they did not show the presence of contractures. Statistically significant changes were observed in the non-sitters group regarding the difference in scale scores and the presence of contractures at the hip and knee. In children with contractures, the improvement in function measured by the scale was lower compared to those without contractures. The sitters group showed that the absence of contractures in both the right and left knee joints resulted in greater improvement in function measured by the Hammersmith scale. In addition, in this group, there were no children with limitations only with one hip joint. Comparison between children with or without contractions in both hips showed a better improvement in children without movement limitations.

In addition, the relationship (correlation using Spearman’s test) between the measured parameters and the magnitude of the difference of the CHOP-INTEND and HFMSE scales scores was examined (Table 5). The non-sitters group showed that the greater the range of left hip flexion (HF-L), the better the CHOP-INTEND score, the smaller the angle of upper trunk rotation (SATR-U), the greater the hip extension (HE) and the smaller the angle of pelvic tilt (PO-ANGLE). The sitters group showed that the greater the range of motion in the hip (HF-L and HF-R), the better the HFMSE score, the lower the angle of upper and lower back rotation (SATR-U and SART-L), the greater the hip extension (HE) and the lower the angle of pelvic tilt (PO-ANGLE). This group presents more statistically significant results, but nonetheless follows a similar pattern.

## 4. Discussion

The current study clearly depicts that the classification of children with SMA is not consistent with the previously proposed types, depending solely on the genetic picture [31]. Patients who developed symptoms before the age of 6 months reach new milestones such as sitting, standing and even walking as a result of treatment, which until recently were only achievable for children with SMA2 and SMA3 [32]. The differences are also seen in adolescents, which could improve or stabilize motor function because of the treatment [12,15].

The group described in the above studies comprised children from a single ethnic and cultural area, treated with modern drug or gene therapy. Their functional level is better than usually reported in the literature, including that of children with the more severe form of spinal muscular atrophy (SMA1). Therefore, there was a need to adjust therapy to not only serve palliative purposes, but to attempt to achieve motor progress. In turn, to select an appropriate therapy, it is necessary to correctly assess the functional status of a child, which motivates research aimed at multifactorial evaluation of posture, structure and functioning of the musculoskeletal system.

Double testing with a functional scale appropriate to the motor level (and not necessarily to the genetic type of SMA) showed that progress appeared in most children. However, the overall improvement in score did not result from all of the items assessed, with only some showing a significant change. It is important that for the first time we confirm that the improvement is not related to all motor skills, rather to selected functions.

It is worth noting that gross motor function in patients undergoing pharmacological treatment improved significantly less, by only one point, in the non-sitters group, compared to the sitters group, where improvements of up to four (first part) and six points (global score) were noted. This result was associated with very distinct improvement especially in two patients diagnosed in the first month of life. The first patient got treatment during our first examination at the age of 1 month and improved by 13 points in functions such as long sitting, one/two hands to head in sitting, propping on forearms, lying to sitting, lifting head from supine, right and left hip flexion in supine. The second patient at the first examination was aged 2 years, but also was diagnosed and received treatment in the first months of life. After 5 months, follow-up showed improvement in function such as lifting head from a prone position, four-point kneeling, crawling, sitting to lying, lying to sitting and high kneeling to half kneeling.

The described changes occurring during the functional development of patients with spinal muscular atrophy have been previously studied. Mercuri et al. demonstrated, at a 12-month follow-up in untreated patients, that HFMSE motor development scores improved by ±2 points. They also observed that the possibility of improvement is higher among children under 5 years of age [33]. Comparing a natural history of SMA with treated patients, it is worth quoting the publication by Pera et al., who observed an increase in HFMSE scores in 75% of 144 patients undergoing pharmacological treatment [34]. In addition, we realize that number of six points of improvement in the sitters group is very impressive, but we suppose to underline that, our study included patients diagnosed after introducing the screening program in Poland. They did not have as many symptoms and the pharmacological treatment has been implemented very fast. It is worth observing that our patients with the highest improvement who were described above were diagnosed and treated in their first months of life. This is supported by the publication of Tscherter et al., where the authors underline that the best improvement results were observed in patients treated before this age [35].

Regardless of effective pharmacological treatment and gene therapy, children with SMA are functioning higher and the correlation with gene copy number seems to be no longer as significant as once thought. We observed in the literature that this lines up with the findings of Pane et al., Pechmann et al. and Tscherter: they did not find a correlation between the SMN2 copy number and motor improvement [29,35,36]. In addition, this study is not about describing the clinical condition but only the functional changes. It is worth noticing that our main aim of the study was a physiotherapeutic analysis of functional and structural changes in patients with SMA undergoing pharmacological treatment and physiotherapy. We were not able to show differences related to the type of drug used for the treatment. In our opinion, both drugs used in this group of patients have similar therapeutic effect.

None of the previous publications provide information on which scale items improve the most, a factor which seems to be very important in view of the physiotherapy provided. It is also noteworthy that most parameters remain unchanged during the five-month follow-up in both groups, which is considered clinically favorable for SMA patients undergoing pharmacological treatment. In the data measured using the goniometer, a slight increase of flexion in motion range was observed in both hips and the left knee in all subjects, while plurimeter measurements showed a worsening of hip extension but an improvement in cervical spine rotation. This is in agreement with the research conducted by Fujak et al., who noted that 96% of SMA patients lose their hip flexion/extension motion range, as well as Stępień et al., who highlighted that this change affected 86% of the participants in their study [20,37]. Salazar et al., on the other hand, showed that even slight range limitations in the hip and knee could affect the motor skills of children with SMA, a finding also confirmed by Wang et al., who stated that contractures in SMA patients were a significant problem, especially for those who lost their ability to walk [2,38]. This information seems to be crucial regarding the extent of physiotherapy provided. The demonstration that the hip joints, in particular, are the most prone to contracture formation may influence the setting of therapeutic goals and physioprophylaxis.

In the following parts of our study, we demonstrated that normal hip range of motion correlates with other parameters. It was noted that the greater the range of motion in the hip (HF), the better the score on the scale, the smaller the angle of upper trunk rotation (SATR-U), the greater the hip extension (HE) and the smaller the angle of pelvic tilt (PO). This partially concurs with the results of Stępień et al., who noted a correlation between muscle strength, thoracic deformities, and angled pelvic positioning [28]. However, the issue itself needs further monitoring, as also confirmed by other authors [4,28,39].

The next step in our study was to examine if joint contracture measured in the supination/exit position affects the difference in scores achieved on the child’s respective scale. It was shown that in the non-sitters group, i.e., in patients with reduced mobility as measured by the CHOP-INTEND scale, the presence of contracture in the hip and knee joint adversely affected the improvement in function. In the sitters group, comprising patients assessed with the HFMSE scale, knee joint contracture had a negative effect on functional improvement. Our findings support those of Nelson et al., who stated that fixed flexion contracture of the hip joint limits, among other things, the functions of turning and lying on the stomach. Similar observations were also demonstrated by Stępień et al. [28,40].

It should be emphasized that the change noted by examining children with the appropriate scale depended largely on the state of the musculoskeletal system, primarily on the position of the pelvis and the range of motion in the hip joints. The presence of contracture in these joints reduces the range of flexion achieved, and thus the ability to make progress in other aspects of pelvic and lower extremity movement. This is illustrated in Table 4: the presence of hip and knee contractures reduces the chance of improvement in the appropriate scale scores. Therefore, resolution of hip contracture appears to be a worthy aspect of functional assessment, and a therapeutic goal worthy of recommendation.

Due to the fact that our article is a pilot single center study, we are anxious to determine whether structural changes have occurred in these children which could affect their functional level. We did not consider the treatment and effectiveness of the physiotherapy. We are aware that treatment is the absolute prerequisite for functional improvement, with physiotherapy as a sustained effect. However, we are trying to show that improvement does not involve the whole motor function, but only selected functions and it is dependent on the presence of contractures.

For the first time during the study, a collective analysis of patients with SMA1, SMA2 and SMA3 was performed. Our study has some limitations. In this article, the different types of SMA were compared. Because spinal muscular atrophy is a very rare disease, we had a highly heterogenous group in terms of SMA type, *SMN2* copy number and age. This may also have been influenced by the fact that the study is of a single center, which may have limited access to patients. For that reason, we could not consider these parameters in the analysis due to insufficient group size. In future studies, a longer follow-up duration should be included. In addition, it would be necessary to expand this study to include a larger number of cases.

## 5. Conclusions

At the five-month follow-up in children undergoing pharmacological treatment and physiotherapy, there was a slight improvement in functional development in the non-sitters group, and a more notable one in the sitters group. In addition, it was noted that the presence of contractures in the hip (worsened in both groups) and knee joint negatively affects functional improvement expressed in scale scores. Therefore, the functional physiotherapy assessment should not be limited to motor scales but also include structural parameters, often disturbed in this group of patients.

## Figures and Tables

**Figure 1 biomedicines-10-03187-f001:**
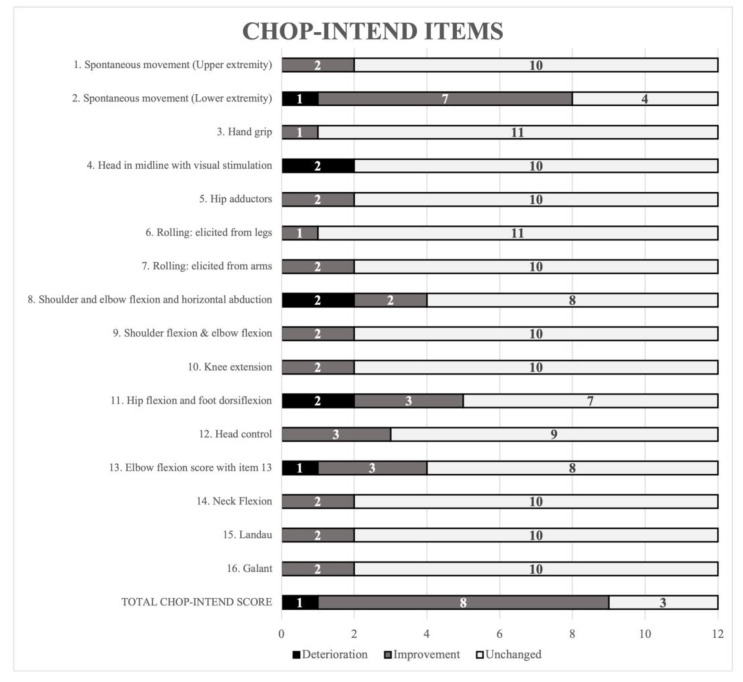
Number of children with deterioration, improvement or unchanged in individual CHOP-INTEND scale items.

**Figure 2 biomedicines-10-03187-f002:**
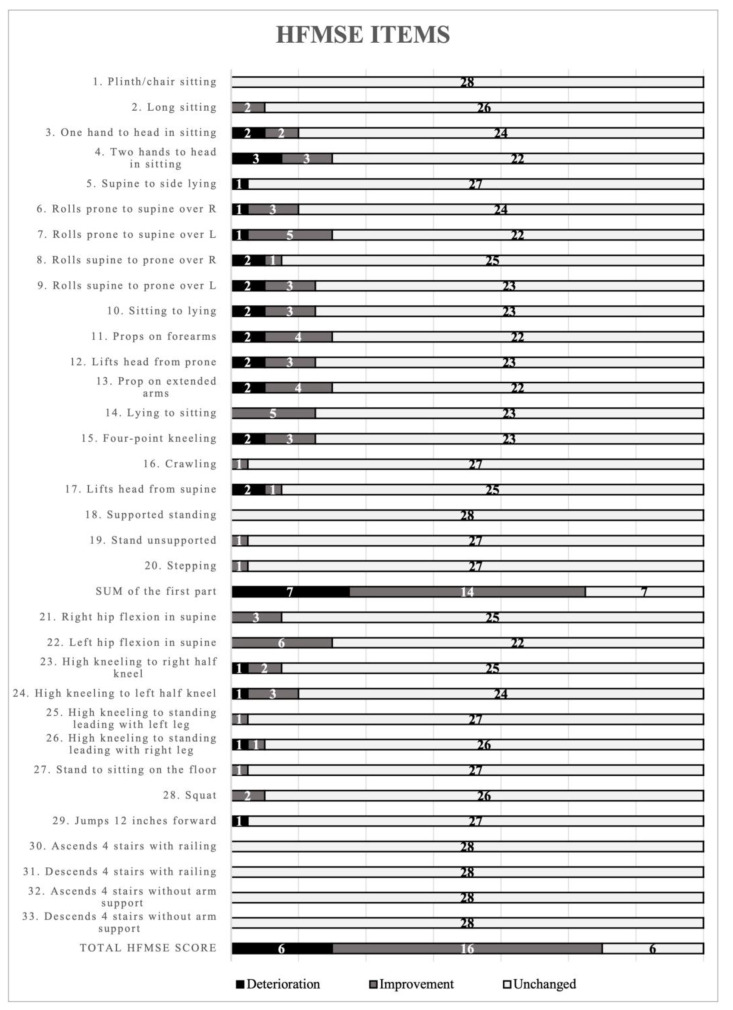
Number of children with deterioration, improvement or unchanged in individual HFMSE scale items.

**Table 1 biomedicines-10-03187-t001:** Study group characteristics.

Parameters	All Participants *n* = 40	Non-Sitters Group *n* = 12	Sitters Group *n* = 28
Age (years): median Q1–Q3	7.50 5.00–12.00	7.50 2.00–12.25	7.50 5.00–12.00
Age at the start of treatment (years): median Q1–Q3	5.00 3.00–11.00	6.00 0.00–10.25	5.00 3.00–11.00
sex: female male	22 18	7 5	15 13
SMA type: SMA1 SMA2 SMA3	14 18 8	12 - -	2 18 8
*SMN2* copy number: 2 copies 3 copies 4 copies	13 21 6	10 1 1	3 20 5
therapy used: Nusinersen Risdiplam Gene therapy	19 19 2	3 7 2	18 10 -
child’s functional level recumbent sits unassisted rolls over crawls stands with assistance stands unassisted walks with assistance walks unassisted	9 8 9 4 3 2 1 4	9 2 1 - - - - -	- 6 8 4 3 2 1 4
rehabilitation intensity: 1–3 times a week 4–6 times a week every day	6 17 17	1 5 6	5 12 11
the duration between the start of treatment to the first examination (months): median Q1–Q3	15 7–32	13 6–21	18 10–36
the duration between the diagnosis to the start of the treatment (months): median Q1–Q3	48 25–111	78 1–122	48 27–97
follow-up duration median	5 months	5 months	5 months

**Table 2 biomedicines-10-03187-t002:** Changes in mobility range parameters between the 1st and 2nd examination. Me (Q25–Q75) and min–max values were indicated. Significance of differences was tested using the sign test, with z- and *p*-values provided.

Parameter		Total *n* = 40	Non-Sitters *n* = 12	Sitters *n* = 28
Goniometer measurement				
HF-L	Examination I	123 (113–130); 100–140	123 (110–128); 100–130	123 (118–133); 105–140
	Examination II	130 (115–130); 95–145	130 (115–130); 95–140	130 (115–133); 105–145
	significance	z = 1.47 *p* = 0.142	z = 1.27 *p* = 0.203	z = 0.80 *p* = 0.421
HF-R	Examination I	120 (112–130); 90–145	115 (105–120); 90–125	120 (115–133); 105–145
	Examination II	128 (110–133); 90–145	123 (105–130); 90–140	128 (113–138); 90–145
	significance	z = 1.65 *p* = 0.098	z = 1.53 *p* = 0.126	z = 1.29 *p* = 0.196
KF-L	Examination I	145 (138–153); 110–160	140 (130–143); 110–155	148 (140–155); 120–160)
	Examination II	145 (140–155); 115–160	140 (130–148); 115–155	150 (140–155); 120–160
	significance	z = 2.13 *p* = 0.033	z = 1.86 *p* = 0.063	z = 1.29 *p* = 0.196
KF-R	Examination I	150 (138–153); 120–160	138 (130–148); 120–150	150 (140–155); 120–160
	Examination II	150 (140–155); 110–160	140 (130–150); 110–155	150 (140–155); 120–160)
	significance	z = 0.08 *p* = 0.936	z = 0.71 *p* = 0.477	z = 0.76 *p* = 0.445
Scoliometer measurement				
SATR-U	Examination I Examination II significance	5 (5–5); 0–15 5 (0–5); 0–15 z = 1.48 *p* = 0.139	5 (5–8); 5–15 5 (5–8); 0–15 z = 0.53 *p* = 0.593	5 (0–5) 0–10 3 (0–5); 0–10 z = 1.47 *p* = 0.142
SATR-L	Examination I Examination II significance	5 (3–8); 0–20 5 (0–5); 0–20 z = 2.93 *p* = 0.003	8 (5–10); 5–20 5 (3–8); 0–20 z = 2.37 *p* = 0.018	5 (0–5); 0–15 5 (0–5); 0–10 z = 1.83 *p* = 0.068
PO-ANGLE	Examination I Examination II significance	5 (5–10); 0–30 5 (5–10); 0–30 z = 0.63 *p* = 0.529	5 (5–5); 0–20 5 (3–5); 0–5 z = 0.00 *p* = 1.000	5 (5–10); 0–30 5 (5–13); 0–30 z = 0.73 *p* = 0.463
Plurimeter measurement				
CR-L	Examination I	78 (70–80); 40–90	70 (58–85); 40–90	80 (70–80); 40–90
	Examination II	80 (70–85); 40–90	80 (60–85); 40–90	80 (70–85); 50–90
	significance	z = 1.95 *p* = 0.051	z = 1.26 *p* = 0.208	z = 1.53 *p* = 0.126
CR-R	Examination I	70 (60–80); 20–85	70 (50–80); 20–85	70 (60–78); 30–85
	Examination II	70 (68–80); 15–90	70 (63–80); 15–85	73 (70–80); 40–90
	significance	z = 2.49 *p* = 0.013	z = 0.28 *p* = 0.779	z = 2.66 *p* = 0.008
HE-L	Examination I	15 (−10–30) −60–45	13 (0–28); −30–40	15 (−10–30); −60–45
	Examination II	10 (−5–25); −45–40	10 (−3–25); −25–40	10 (−8–28); −45–40
	significance	z = 1.12 *p* = 0.264	z = 1.13 *p* = 0.260	z = 0.64 *p* = 0.520
HE-R	Examination I	15 (−3–30) −50–45	13 (0–33); −20–35	18 (−5–28); −50–45
	Examination II	10 (−10–30); −40–45	10 (−8–28); −30–40	10 (−13–30); −40–45
	significance	z = 0.10 *p* = 0.318	z = 0.42 *p* = 0.674	z = 0.88 *p* = 0.378

Abbreviations: HF-L left hip flexion range; HF-R right hip flexion range; KF-L left knee flexion range; KF-R right knee flexion range; SATR-U upper trunk rotation angle; SATR-L lower trunk rotation angle; PO-Angle pelvic angle; CR-L left cervical rotation; CR-R right cervical rotation; HE-L left hip extension; HE-R right hip extension; statistical significance *p* < 0.05.

**Table 3 biomedicines-10-03187-t003:** The changes in the structural assessment of the musculoskeletal system, divided according to the scale used for the study. The numbers of children in whom the baseline value was correct and did not worsen (normal range), a previously abnormal range improved, the range remained abnormal (no change), or a worsening occurred, were all reported.

Parameters	All Participants *n* = 40				Non-Sitters Group *n* = 12				Sitters Group *n* = 28			
	Normal Range	Improvement	No Change	Worsening	Normal Range	Improvement	No Change	Worsening	Normal Range	Improvement	No Change	Worsening
HF-L	8	**18**	3	11	2	**7**	0	3	6	**11**	3	8
HF-R	4	**20**	3	13	0	**8**	0	4	4	**12**	3	9
KF-L	7	**17**	10	6	1	**6**	4	1	6	**11**	6	5
KF-R	7	11	**14**	8	0	**6**	3	3	7	5	**11**	5
SATR-U	10	7	**21**	2	0	2	**9**	1	10	5	**12**	1
SATR-L	11	11	**18**	0	0	**7**	5	0	11	4	**13**	0
PO-ANGLE	5	3	**19**	5	0	1	**2**	1	5	2	**17**	4
CR-L	4	**18**	9	9	3	**4**	3	2	1	**14**	6	7
CR-R	0	**18**	11	11	0	3	4	**5**	0	**15**	7	6
HE-L	1	12	6	**21**	1	3	2	**6**	0	9	4	**15**
HE-R	1	14	8	**17**	1	4	3	4	0	10	5	**13**

Abbreviations: HF-L left hip flexion range; HF-R right hip flexion range; KF-L left knee flexion range; KF-R right knee flexion range; CR-L left cervical rotation; CR-R right cervical rotation; HE-L left hip extension; HE-R right hip extension. The most relevant information has been bolded.

**Table 4 biomedicines-10-03187-t004:** Comparison of the difference in score in the respective scales, in relation to the presence of contracture in the initial joint position at the first examination. The Mann–Whitney U test was used to assess the significance of the difference between the subgroups, whereas the sign test was used to assess the difference in time.

**Non-Sitters Group, *n* = 12**						
Contracture, examination I	Both hips, both knees, YES, *n* = 10			Both hips, both knees, NO, *n* = 2		
CHOP I SUM	15 (2–37)			32; 52		
CHOP II SUM	16 (5–35)			59; 60		
difference between examinations I and II	−4 to +6			−8 to 27		
Significance of the difference between the I and II tests, in relation to the presence of contracture	**z = −2.06; *p* = 0.039**					
**Sitters group, *n* = 28**						
contracture, examination I	Both hips	Both hips	Left knee		Right knee	
	YES, *n* = 18	NO, *n* = 10	YES, *n* = 22	NO, *n* = 6	YES, *n* = 23	NO, *n* = 5
HFMSE I SUM	18 (5–54)	39 (25–59)	22 (14–35)	33 (25–61)	23 (14–36)	26 (25–41)
HFMSE II SUM	20 (6–54)	38 (30–59)	25 (14–35)	39 (29–63)	25 (14–38)	34 (29–44)
difference between examinations I and II	−4 to +9	−4 to +13	−4 to +13	+1 to +9	−4 to +13	+1 to +9
Significance of the difference between the I and II tests, in relation to the presence of contracture	**z = −1.65** ***p* = 0.010**		**z = 2.18** ***p* = 0.030**		**z = 2.12** ***p* = 0.034**	

**Table 5 biomedicines-10-03187-t005:** The relationship (Spearman’s test correlation) between the measured parameters and the magnitude of the difference of the CHOP and HFMSE scale scores.

Parameters	Rho Value=
**Non-Sitters Group**	
HF-L and total scale score, examination II	0.736
HF-L and SATR-U, examination II	−0.580
HF-L and HE-L, examination II	0.588
HF-L and PO-ANGLE, examination I	−0.668
HF-R and PO-ANGLE, examination I	−0.682
**Sitters group**	
HF-L and total scale score, examination II	0.528
HF-R and total scale score, examination II	0.581
HF-L and CR-R, examination II	0.427
HF-L and SATR-U, examination I	−0.460
HF-R and SATR-U, examination I	−0.387
HF-L and SATR-L, examination I	−0.424
HF-R a SATR-L, examination I	−0.407
HF-L and HE-L, examination II	0.413
HF-R and HE-L, examination II	0.433
HF-L and HE-R, examination II	0.433
HF-R and HE-R, examination II	0.435
HF-R and PO-ANGLE, examination I	−0.384
HF-L and PO-ANGLE, examination II	−0.498
HF-R and PO-ANGLE, examination II	−0.0534

## Data Availability

Not applicable.

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
