# Peer review of "Prospective Analysis of Functional and Structural Changes in Patients with Spinal Muscular Atrophy—A Pilot Study"

_biomedicines, 2022, doi:10.3390/biomedicines10123187_

Round 1

Reviewer 1 Report

Bieniaszewska et al present a prospective single-centre cohort examining the association between motor progression and structural changes in treated SMA patients. Major revisions are required:
At the cohort both children and adolescents up to 20 years old are included. Nevertheless at the introduction authors only cite papers about treatment effectiveness in infants and children. A comment about treament effect on adolescents should be made (PMID: 35837793, PMID: 35178673).
Authors should explain the prospective design of this study. Were the data prospectively collected? Were all consecutive patients examined at the clinic included in this study?
At the introduction the limitations of the scales used in clinical practice should be described (
PMID: 35115230).
The study has some more limitations than those described: the fact that is single centre and the limited follow-up.
Discussion section is extensive and not well-written. It should be revised in order to give some clear messages.
The fact that the patients with the highest improvement were diagnosed and treated earlier should be clear at the results. Disease duration before treatment initiation should be presented at the table along with the p of the comparison between the 2 groups. Considering this the conclusion phrase : "At the five-month follow-up in children undergoing pharmacological treatment and  physiotherapy, there was a slight improvement in functional development in the non-sitters group, and a more notable one in the sitters group." should be revised accordingly.

Reviewer 2 Report

In this paper Bieniaszewska and colleagues aim to address functional and structural changes in children with SMA undergoing farmacologic or genetic treatment and physiotherapy in a prospective study.

Beside the fact that the aim is interesting, the paper is too less informative to provide scientific hints and to reach consistent conclusions. In the Introduction, some short information about the main pharmacologic targets and mechanism of action of the drugs currently used for therapeutic approaches should be included. The major results about their effects known so far should also be summarized. A brief description about the pathophysiology of SMA should be provided as well.  No description about the type and the protocol of physiotherapy is available. The results are not reported in association to the type of drugs used for the treatment (Nusinersen or Risdiplam). If stratified by the type of treatment the number of subjects probably will be too small to perform statistical analyses, therefore total number of enrolled patients should be increased. Did the Authors think that the improvements are the results of physiotherapy or of the drugs? or of a combination of both? 

In Figure 1 and 2 it is not clear how the authors calculated the total sum of numbers. The results provided in numbers in the Tables, would be more readable in a graph form. 

Round 2

Reviewer 1 Report

The manuscript was improved following reviewer's suggestions.

An extensive editing of English language and style by a native speaker are required.

The data about patients baseline characteristics should be removed from paragraph 2.1 Materials. They should be presented at results.

The discussion still needs improvement. The importance of the findings and their translation into clinical practice should be clearly presented.

The conclusions should not be presented in a bullet form.

Reviewer 2 Report

The authors addressed the raised questions. Please read my comments in blue in the attached file.

Round 3

Reviewer 2 Report

I'm fine with the revision. Thank you.